# Striped nanoscale phase separation at the metal–insulator transition of heteroepitaxial nickelates

G. Mattoni[1], P. Zubko[2], F. Maccherozzi[3], A.J.H. van der Torren[4], D.B. Boltje[4], M. Hadjimichael[2], N. Manca[1], S. Catalano[5], M. Gibert[5], Y. Liu[3], J. Aarts[4], J.-M. Triscone[5], S.S. Dhesi[3] & A.D. Caviglia[1]

Nucleation processes of mixed-phase states are an intrinsic characteristic of first-order phase transitions, typically related to local symmetry breaking. Direct observation of emerging mixed-phase regions in materials showing a first-order metal–insulator transition (MIT) offers unique opportunities to uncover their driving mechanism. Using photoemission electron microscopy, we image the nanoscale formation and growth of insulating domains across the temperature-driven MIT in $NdNiO_3$ epitaxial thin films. Heteroepitaxy is found to strongly determine the nanoscale nature of the phase transition, inducing preferential formation of striped domains along the terraces of atomically flat stepped surfaces. We show that the distribution of transition temperatures is a local property, set by surface morphology and stable across multiple temperature cycles. Our data provide new insights into the MIT of heteroepitaxial nickelates and point to a rich, nanoscale phenomenology in this strongly correlated material.

[1] Kavli Institute of Nanoscience, Delft University of Technology, 2628 CJ Delft, Netherlands. [2] London Centre for Nanotechnology and Department of Physics and Astronomy, University College London, 17–19 Gordon Street, London WC1H 0HA, UK. [3] Diamond Light Source, Harwell Science and Innovation Campus, Chilton OX11 0DE, UK. [4] Kamerlingh Onnes-Huygens Laboratory, Leiden University, P.O. Box 9504, 2300 RA Leiden, Netherlands. [5] Département de Physique de la Matière Quantique, University of Geneva, 24 Quai Ernest-Ansermet, 1211 Genève 4, Switzerland. Correspondence and requests for materials should be addressed to G.M. (email: g.mattoni@tudelft.nl).

Rare-earth nickelates are strongly correlated electron systems in which structural and electronic properties are interconnected[1,2]. A well-studied member of this family is NdNiO₃, which shows a first-order temperature-driven metal–insulator transition (MIT) accompanied by a structural phase change and the appearance of unconventional magnetic order[3–6]. Several models have been proposed to describe its electronic structure, however the microscopic mechanism of the phase transition is still debated[7–13]. A number of experiments underscores the key role of the lattice, as demonstrated by the influence of hydrostatic pressure, epitaxial strain and resonant phonon excitation on the MIT[14–23]. The coexistence of metallic and insulating regions in the vicinity of the MIT, typical of first-order phase transitions, has been discussed with an expected domain size of a few tens of nanometre[16,24]. However, the formation of insulating domains has been inferred, so far, mainly from macroscopic transport measurements[25–28]. In this thermodynamic limit the influence of nanoscale control parameters, such as local strain fields, lattice distortions and inhomogeneity, is buried in the statistical average of multiple domains. To achieve fundamental understanding and control of phase separation, access to the nanoscale regime is required[29,30].

Several methodologies have been used for nanoscale imaging of mixed metallic and insulating phases in correlated oxides, including scanning tunnelling microscopy[31], near-field infrared microscopy[32] and scanning electron microscopy[33]. Nanoscale phase separation across a phase transition has also been studied using photoemission electron microscopy (PEEM)[34,35].

Here we use PEEM to image nano-domain formation and disappearance in NdNiO₃. This technique combines a spatial resolution of a few tens of nanometres with real-time imaging, allowing us to track the MIT in nickelates at different stages of its evolution. Our findings show that heteroepitaxy of NdNiO₃ on atomically flat stepped surfaces leads to the formation of striped insulating domains, which nucleate and grow along surface terraces across the MIT. We discuss how morphological characteristics act as a template for phase separation, determining the local transition temperature, as well as domain nucleation and growth pathways. Our data provide evidence, for the first time in the nanometre range, for the strong coupling between structural and electronic degrees of freedom in the rare-earth nickelates.

## Results

**Sample characterization.** For this experiment a 30-unit-cell-thick NdNiO₃ (001)$_{pc}$ epitaxial film was grown on a NdGaO₃ substrate. The epitaxial strain imposed by the substrate sets the transition temperature for the MIT and the width of the hysteresis loop[17]. As shown in the atomic force microscopy image and related line profile in Fig. 1a,b, the film presents an atomically flat surface with steps and terraces that mimic the underlying substrate. Figure 1c shows a $\theta - 2\theta$ X-ray diffraction scan around the (001)$_{pc}$ peak of the NdNiO₃ film. Finite size oscillations are observed and fitted with a kinematic scattering model, indicating high crystalline quality and confirming the expected film thickness of about 11 nm.

The MIT hysteresis, associated with the formation of insulating domains, is measured by four probe d.c. transport (Fig. 1d). We define the transition temperatures $T_{MI} = 150$ K and $T_{IM} = 178$ K as the peaks of $-\partial \log R/\partial T$ on cooling and heating, respectively (Supplementary Fig. 1). From the peaks separation we calculate the hysteresis width $\Delta T_{MIT} = 28$ K. In agreement with previous reports[16], the MIT width in thin films appears much broader than in bulk NdNiO₃, signalling the influence of heteroepitaxy on the phase transition evolution.

**Imaging contrast mechanism.** To image the different electronic phases, we perform X-ray absorption spectroscopy (XAS) at Ni L₃

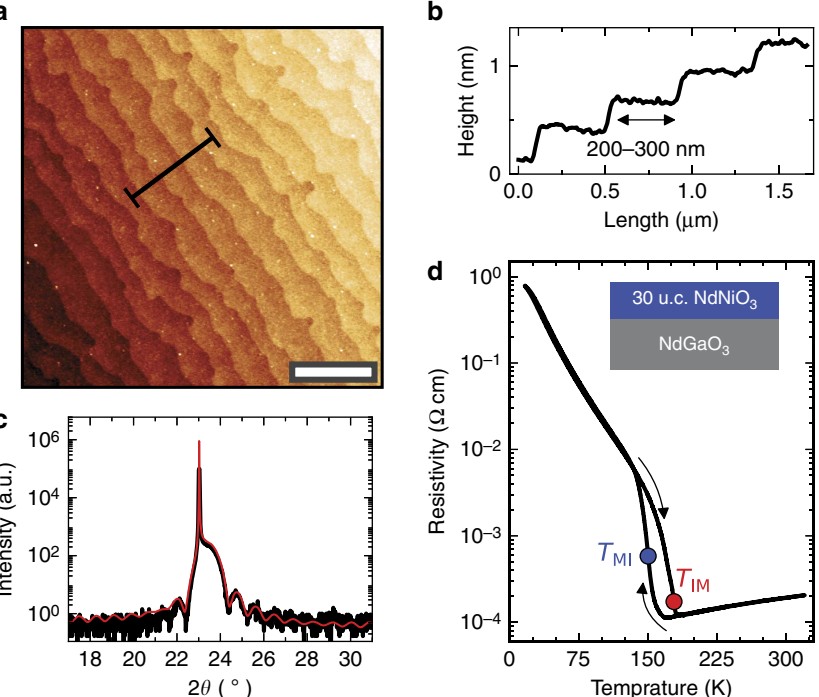

**Figure 1 | Characterization of the 30-unit cell-thick NdNiO₃ film.** (**a**) Surface morphology with single-unit cell steps (~0.4 nm) and terraces (200–300 nm in width) as measured by atomic force microscopy; scale bar, 1 μm. (**b**) Cross-section profile showing the film step height and average terrace width. (**c**) X-ray diffraction data (black) around NdNiO₃ (001)$_{pc}$ peak, fitted with a kinematic scattering model (red). (**d**) Resistance versus temperature from the transport measurement, where $T_{MI}$ (blue dot) and $T_{IM}$ (red dot) are indicated as extracted from the peaks of $-\partial \log R/\partial T$. (u.c., unit cell).

absorption edge. We use $\sigma$-polarized X-rays and acquire the signal in total electron yield, thus probing the material surface down to a few nanometres. In Fig. 2a the temperature dependence of the Ni $L_3$ XAS is presented (see Supplementary Fig. 2 for XAS on a broader photon energy range and with different polarization). The most intense absorption peak shifts towards lower photon energies upon cooling the sample from the metallic state at $T = 185$ K to the insulating one at $T = 140$ K. This is consistent with an increased energy splitting of the Ni $L_3$ multiplet in the insulating phase due to a partial change of Ni valence state[13,36]. The observed energy shift provides a contrast mechanism suitable for our study.

Above the MIT, the XAS spectra measured over the full field of view do not display significant variations compared with the noise level of the experiment. Below the MIT, instead, the sample shows different spatially dependent XAS spectra, divided in two subsets with a relative shift in absorption edge (Fig. 2c). The maximum difference between the two subsets spectra is observed at 852.0 and 852.7 eV. We thus construct electronic phase maps by acquiring PEEM images at these two photon energies, and calculating their difference pixel by pixel. The use of fixed energy values slightly reduces the PEEM contrast in the high-temperature region, but allowed us to perform faster acquisitions, thus increasing the number of data points taken during the temperature ramps (for further details see Supplementary Fig. 3 and Supplementary Note 1). In all images a round-shaped surface defect (dashed square) provides a well-contrasted reference feature used to compensate for the time-dependent spatial drift and keep the same area of interest in focus during the experiment.

At $T = 185$ K the resulting map (Fig. 2b) is spatially homogeneous, while at $T = 140$ K alternating bright and dark features (Fig. 2d) appear. We identify the bright features as insulating domains nucleating in a metallic matrix during the MIT. Indeed, as shown in Fig. 2c, the bright features display local spectra that are shifted to lower energies when compared with the dark ones. Such shift is in qualitative agreement with the spatially averaged XAS spectra of Fig. 2a measured above and below the transition temperature. We note that even if the detection of this energy shift is at the resolution limit of the PEEM technique, we obtained a sufficiently high signal-to-noise ratio by considering relative intensity differences at two distinct energies.

If we compare the PEEM and atomic force microscopy measurements acquired with the same sample orientation (Figs 1a and 2d), we find a direct relationship between the insulating domains and surface morphology. Our *ex situ* comparison is allowed by the single-crystal nature of our samples (see additional X-ray diffraction characterization in Supplementary Fig. 4), where the surface terraces orientation is preserved over millimetres. The surface terraces act as nucleation centres for the insulating phase, so that insulating domains form and grow preferentially along them, resulting in a striped shape. This is consistent with reports of sensitivity to strain for the nickelates[17], suggesting that the local periodic strain field at the step edges can confine the insulating phase on the terraces, limiting its expansion. These observations have been reproduced on three different samples, confirming that the domain orientation and size are dictated by surface morphology (Supplementary Fig. 5). Our finding establishes an important link between sample local morphology and electronic phase separations.

**Nanoscale evolution of the MIT.** To investigate the evolution of the insulating domains across the MIT, the sample temperature is cycled below and above the transition, following the hysteresis loop. A representative set of images is reported in Fig. 3a. The percentage of area covered by the insulating domains in each PEEM image of the series is presented in the inner panel of Fig. 3a. From room temperature down to 152 K the sample shows

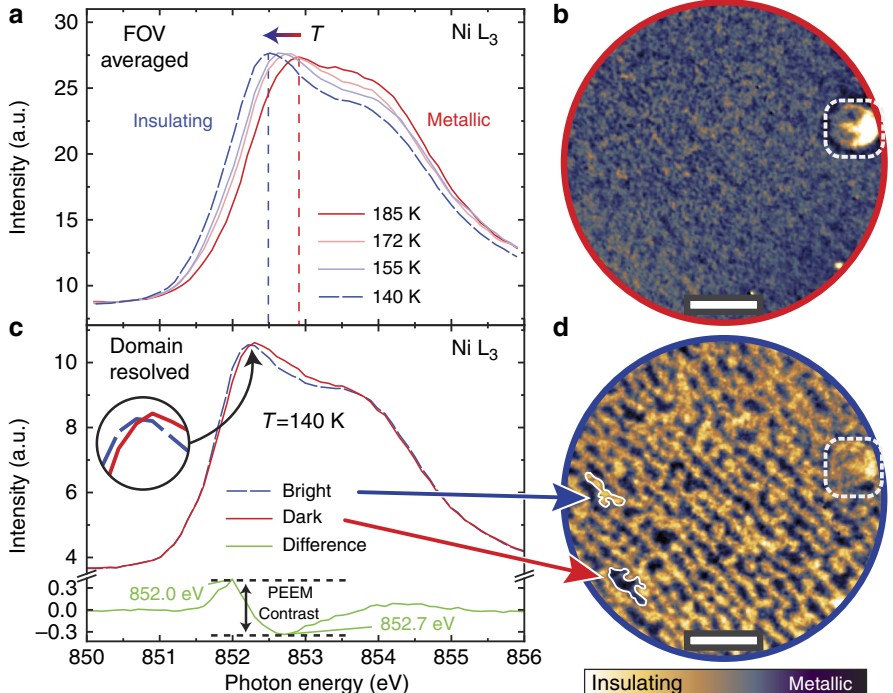

**Figure 2 | The PEEM imaging contrast from photon-energy-shifted XAS spectra of metallic and insulating phases.** (**a**) Temperature dependence of Ni $L_3$ XAS spectra measured over the full field of view. PEEM images showing (**b**) the metallic phase at 185 K and (**d**) the insulating phase at 140 K. A surface defect used as a reference feature for drift correction is indicated by the dashed square. (**c**) Domain-resolved XAS spectra of bright and dark features in **d**. Scale bar, 1 μm.

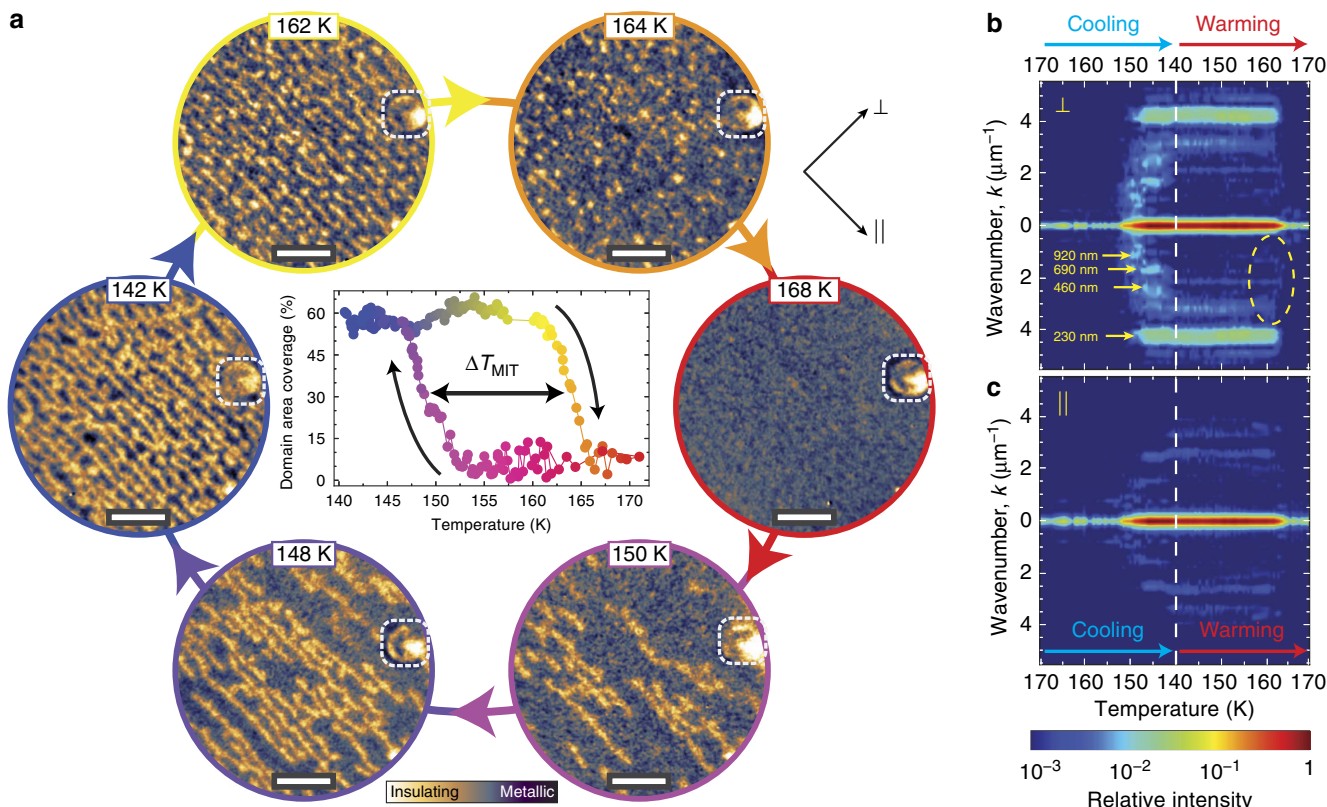

**Figure 3 | Temperature evolution of insulating domains across the MIT.** (**a**) During each thermal cycle the insulating domains nucleate and grow on cooling, while they gradually disappear on warming (see video in the Supplementary Material). The inner panel shows the percentage of image area covered by the insulating domains as a function of temperature, highlighting the hysteretic behaviour of the MIT with a finite width down to the single domain. Scale bar, 1 μm. (**b**) Perpendicular and (**c**) parallel linecuts of the two-dimensional Fourier transform with respect to the insulating domain orientation (indicated by the black arrows) as a function of temperature. The colour scale represents the power spectrum normalized with respect to the maximum value at $T = 140$ K and $k = 0$. The dashed ellipse evidences the asymmetry between the cooling and warming directions.

a homogeneous metallic phase. Below 152 K insulating domains nucleate and grow along the preferential direction given by surface terraces, gradually forming striped regions. Between 146 and 140 K the domain evolution saturates at about 60% coverage and no additional insulating regions are formed. This domain configuration is stable for the whole duration of the measurements (several hours).

The reverse transition, back to the metallic state, is rather different. On heating, no change is initially observed up to 161 K, in agreement with the hysteretic, first-order nature of NdNiO$_3$ MIT. Above 161 K the insulating stripes become narrower and are pinched off by the expanding metallic matrix into many, small and closely spaced nano-domains. These appear to be evenly distributed across the field of view, in stark contrast to the striped domains observed on cooling (that is, compare the Fig. 3a heating and cooling images at 164 and 150 K, respectively, with approximately the same insulating domain coverage). At $T = 165$ K all the insulating domains disappear and the homogeneous metallic phase is recovered. Interestingly, we note that the insulating domains do not fully populate the surface as the area coverage reaches the saturation value of about 60%. We observe no significant variation of the coverage down to 130 K, the lowest temperature attainable in our experiment. The domains are often spaced by metallic regions, which persist at the surface step edges. This effect might be related either to local strain fields in proximity of the step edges or to inhomogeneous surface termination.

A clear asymmetry between the metal-to-insulator and insulator-to-metal transition is underscored by the two-dimensional Fourier transform (FT) of the PEEM images acquired

during the temperature cycle. The temperature dependence of the FT power spectrum line-cuts along the direction perpendicular and parallel to the striped domains is reported in Fig. 3b,c. The intensity at $k = 0$ corresponds to the domain area coverage in the inner panel of Fig. 3a, and its maximum value at 140 K is used to normalize the spectrum. In the perpendicular direction (Fig. 3b) we see the appearance of an intense peak at the nucleation of the insulating phase, which corresponds to a periodicity of about 230 nm. This value matches with the average terrace width of Fig. 1b, highlighting the direct relationship between insulating domains and surface morphology. On cooling, the domain formation pattern is characterized by the appearance of peaks corresponding to multiple integers of the terrace width. These features disappear when the domain area coverage reaches saturation. Remarkably, these additional peaks are absent during warming (dashed ellipse in Fig. 3b), indicating that a different pattern underlies the disappearance of the insulating phase. The formation of the insulating domains is thus a nucleation and growth process, while their disappearance is a homogeneous melting that originates from the domain edges. This is consistent with previous reports[25], where a supercooling mechanism was associated with the metal-to-insulator transition only. In the parallel direction (Fig. 3c), instead, negligible domain ordering is observed, where dim peaks one order of magnitude weaker than in the perpendicular case appear. We relate this signal with the average on-terrace distance of the residual metallic matrix in the insulating phase.

The presented MIT evolution is consistent across multiple temperature cycles. This allows us to assign local transition

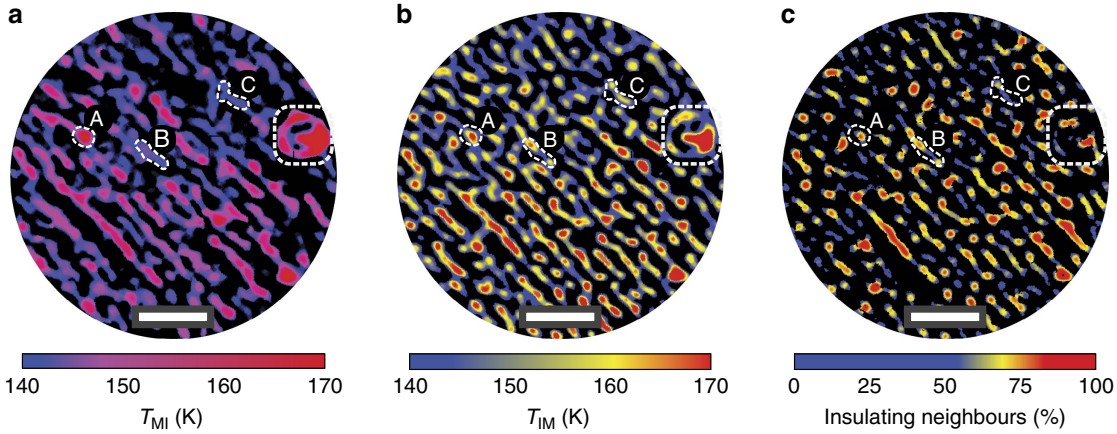

**Figure 4 | Spatial distribution of MIT temperature.** Maps of (**a,b**) $T_{MI}$ and $T_{IM}$ showing the transition temperature is a local property of the material. Three areas are encircled to indicate the order with which they become insulating (A, B and C) and revert to the metallic state (C, B and A). The map in (**c**) shows point-by-point the amount of insulating region at 140 K in a neighbouring area of radius 100 nm. Scale bar, 1 μm.

temperatures to the material. In Fig. 4a,b we present spatially resolved maps of local $T_{MI}$ and $T_{IM}$, showing the temperature at which the phase transition occurs on a certain region of the sample. Repeating the temperature cycle several times, the insulating domains are observed nucleating and growing always in the same position and in the same order. As an example we considered the areas labelled as A, B and C in Fig. 4a,b. If on a cool-down cycle they turn insulating in an (A, B and C) order, during a warm-up they will revert to the metallic state in the reversed (C, B and A) order.

We find in Fig. 4b that the spatial distribution of insulator-to-metal transition temperature seems to be related to the size and shape of the domains themselves. In particular, the cores of bigger domains show higher values of $T_{IM}$. This indicates that the melting process of the insulating phase starts from the domain edges. To support this observation, in Fig. 4c we determine the insulating regions neighbouring each insulating point in a radius of 100 nm at 140 K. We note how the data in Fig. 4c are evaluated from a single PEEM image at 140 K, in contrast to Fig. 4a,b, which are extracted by using all the images in the temperature cycle. The striking similarity between Fig. 4b,c is a clear indication of how the insulator-to-metal transition progresses continuously from the edges to the core of each domain, so that the bigger ones are the last to disappear. This is in agreement with previous reports of an intrinsic asymmetry in the phase transition of NdNiO$_3$ (ref. 25).

A relevant result of our analysis is the preservation of the MIT hysteresis down to the single domain. From the inner panel of Fig. 3a we can extract the hysteresis width $\Delta T_{MIT} = (14 \pm 2)$ K, defined as the temperature difference between appearance and complete melting of the insulating phase. This value is in sharp contrast with our macroscopic transport measurements, where we found $\Delta T_{MIT} = 28$ K. We also note that the observed nanoscale inhomogeneities appear on a smaller length scale than the field of view used in the experiment, thus providing a representative evaluation of the properties of the material. The existence of a finite hysteresis width down to the single domain scale, and the spatial distribution of $T_{MI}$ and $T_{IM}$ provide a further insight on the nature of the phase transition. Such results can hardly be inferred by macroscopic measurements, which are subject to statistical averaging.

**MIT through bulk and surface techniques.** At this point it is worth comparing the temperature dependence of the domain area coverage measured by PEEM with the resistivity data, both shown

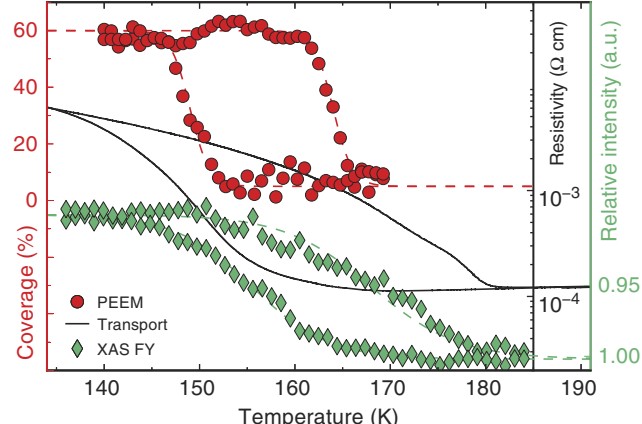

**Figure 5 | MIT hysteresis measured with different techniques.** Insulating domain area coverage from the inset of Fig. 3a (red dots), low-temperature transport from Fig. 1d (black line) and X-ray absorption intensity at 853 eV photon energy relative to the intensity at 180 K measured in fluorescence yield (FY, green diamonds).

in the relevant temperature range in Fig. 5. We see a striking difference in the extremal temperatures of the two hysteresis loops. In the PEEM data the hysteresis loop closes at 165 K on the high-temperature end, about 15 K lower than in the transport case. This means that while the insulating domains coverage goes to zero at 165 K during a warming ramp, the resistivity is still almost an order of magnitude higher than in the metallic state. Assuming the domains propagate completely through the film thickness (that is, that 0% area coverage also corresponds to 0% volume fraction), it is not possible to explain the difference between area coverage and transport.

To get a further insight into this difference, we additionally measure the evolution of macroscopic XAS intensity at 853 eV in the fluorescence yield configuration as a function of temperature (green diamonds in Fig. 5). The MIT hysteresis measured this way is in qualitative agreement with the transport data. In contrast to the measurements in total electron yield performed in the PEEM set-up, the XAS in fluorescence yield probes the whole thickness of our NdNiO$_3$ thin film, also providing a more extended spatial averaging as the X-ray spot size is about two orders of magnitude larger.

We consider two different explanations for the measured discrepancy between PEEM and transport/XAS data. A possibility

is that the observed domains do not fully penetrate through the whole film but are, instead, confined to the surface. In this case, our measurements indicate that the MIT at the surface occurs at a lower temperature than in the bulk, eventually related to local lattice distortions at the free boundary. Another explanation involves the possibility of local material metallization due to X-ray illumination. This is a well-known open issue when irradiating oxide materials with intense X-rays, and changes in metal–insulator characteristics have been previously reported[37]. Since the use of X-rays is an intrinsic requirement of the PEEM technique, further insight into this question will be provided by additional experiments, such as transport measurements at the nanoscale and scanning probe techniques.

## Discussion

Through direct imaging by PEEM, we reported on nanoscale phase separation during the MIT of $NdNiO_3$ thin films. Striped domains nucleate and grow along the terraces of the atomically flat surface, highlighting the influence of heteroepitaxy on the phase transition. Performing systematic imaging as a function of temperature, we showed that the transition temperature is a local property of the material, stable across multiple temperature cycles. The measurements point towards a strong interconnection between structural and electronic degrees of freedom in rare-earth nickelates, suggesting a new approach for controlling phase separation at the nanoscale.

## Methods

**Sample fabrication.** The commercially available $NdGaO_3$ substrate was annealed at 1,000 °C in 1 atm of oxygen before sample growth to achieve a flat surface with a regular step and terrace structure. The $NdNiO_3$ $(001)_{pc}$ film was grown by off-axis radio-frequency magnetron sputtering in $1.80 \times 10^{-1}$ mbar of an oxygen/argon mixture of ratio 1:3 at a substrate temperature of 490 °C.

**Temperature-dependent measurements.** Transport, PEEM and XAS fluorescence yield measurements have been performed cycling sample temperature at a constant rate of 0.5 K min$^{-1}$ for both ramp directions, guaranteeing that the sample is kept in a quasi-static condition. As each PEEM acquisition required about 20 s, we estimate an error of 0.2 K on each data point. Such temperature variation is negligible compared with the phase transition evolution.

**Synchrotron X-ray measurements.** PEEM and XAS fluorescence yield data have been acquired at the beamline I06 of Diamond Light Source. An X-ray beam $10 \times 10$ μm in spot size with a fluency of 1 mJ cm$^{-2}$ was used in the PEEM set-up, while a $100 \times 100$ μm beam with 0.01 mJ cm$^{-2}$ of fluency was used for the XAS fluorescence yield measurements. In both cases the X-rays were σ-polarized. The absolute peak photon energies measured by PEEM are subjected to an uncertainty of about 0.2 eV due to the small integration time of 1 s compared with the noise level of the system. However, this does not affect the reported data as all PEEM images are based on relative spatial shifts of X-ray absorption intensity.

**Data availability.** All relevant data are stored at the ICT facilities of TU Delft and are available from the authors on request.

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

## Acknowledgements

We thank Dejan Davidovikj for useful discussions and suggestions. We acknowledge the Foundation for Fundamental Research on Matter (FOM), the Nanofront consortia and NWO-nano programme funded by the Netherlands Science Foundation NWO/OCW, EPSRC grant number EP/M007073/1, and Diamond Light Source for the

provision of beamtime under proposal numbers SI-13081 and SI-10428. This work was supported by the Swiss National Science Foundation through Division II and the European Research Council under the European Union's Seventh Framework Program (FP7/2001-2013)/ERC Grant Agreement no. 319286 (Q-MAC).

## Author contributions

G.M., P.Z., F.M., A.J.H.T., D.B.B., M.H. and S.S.D. performed the PEEM measurements; G.M. performed the transport and atomic force microscopy measurements; S.C. and M.G. grew the films and performed the X-ray diffraction characterization; S.S.D. and F.M. performed the XAS fluorescence yield measurements; G.M. analysed the data; G.M. and A.D.C. wrote the experiment proposal with contributions from S.S.D.; G.M., N.M. and A.D.C. wrote the manuscript; A.D.C. conceived and supervised the project. All authors reviewed and edited the manuscript.

## Additional information

**Competing financial interests:** The authors declare no competing financial interests.

