## [Peer Review File · Nature Communications]

Reviewers' comments:

Reviewer #1 (Remarks to the Author):

In the manuscript 'Striped nanoscale phase separation at the metal-insulator transition of heteroepitaxial nickelates', the authors have applied PEEM technique around the first order metal-insulator transition of NdNiO₃ thin film. The authors claim to observe nucleation of the insulating phase within metallic matrix by using PEEM.

First, I would to point out that PEEM has been successfully used before to study the phase nucleation process in other systems e.g. VO₂ (arxiv:1503.07892), FeRh arXiv:1405.4319. Moreover, the existence of a mixed electronic phase within the hysteresis region of NdNiO₃ had been demonstrated before (e.g. Journal of Applied Physics 108, 063503 (2010); Journal of Physics: Condensed Matter 21 (48), 485402 (2009); Journal of Physics: Condensed Matter 21 (18), 185402 (2009)). Nevertheless, the application of PEEM to microscopically investigate the nucleation process across the 1st order MIT is new and interesting.

Before the final decision is made the author should address several questions and issues states below:

1. First, the author must add the color scale in Fig. 2(b), (c) and Fig. 3(a)! Otherwise, it is not clear how much the regions are different.
2. The PEEM images are constructed by using the difference of area scan with photon energy of 852.0 and 852.7 eV. It is not clear what microscopic parameter do these two specific energies correspond to? It is well known that the amount of peak splitting (roughly 852.0 eV and 853.7 eV in Fig. 2(c)) can be modeled as a measure of charge transfer energy (e.g. Phys. Rev. B **83**, 161102(R) (2011)). However, as the peak splitting and the relative weight of the two peaks of L3 edge evolve with the temperature, why do the authors map the area with the same energy at all temperatures?
3. The authors claim that the insulating phases nucleate along the terraces on the substrate by performing AFM and PEEM with the same orientation of the sample. However, there is no experimental evidence that the film used for the study is crystallographically a single domain sample. This needs to be explained.
4. Can the author comment why does the insulating phase nucleate along the terraces?
5. Do the author observe any time dependence in PEEM image at a fixed temperature within the hysteresis region. Earlier time dependence measurement demonstrated that Hall coefficient (Appl. Phys. Lett., 103, 182105 (2013)) drifts with time, naturally implying the metallic and insulating regions evolve in time.
6. The width of the hysteresis curves of PEEM is much narrower compared to both the hysteresis of transport data and hysteresis of FY XAS. The authors assign this effect to the surface sensitivity of PEEM. However, the main point of this manuscript is to map out the phase nucleation around the first order metal insulator transition (existence of such nucleation process is already known) as the bulk property - as such this discrepancy must be addressed properly. Can the author also plot the hysteresis from

TEY XAS? One more remark: The author should carry out the PEEM experiments on a sample with the thickness comparable with to the probing depth of PEEM.

Minor issues:

1. Since NdNiO₃ is orthorhombic, the author should explicitly mention that the orientation (001) is in pseudo cubic setting (?)
2. In the introduction, the author mentioned various techniques except for PEEM that have been used to map mixed metallic and insulating phases. As PEEM has been also used to investigate such phenomenon, the author should cite the appropriate PEEM refs, e.g. mentioned earlier in this report.

Before taking the final decision I would like to look at the author response to my comments and the revised text.

Reviewer #2 (Remarks to the Author):

The authors present results of a temperature dependent XAS-PEEM study investigating the spatial evolution of the metal / insulator domain behavior through the MI transition of the rare earth NdNiO₃ film grown on STO(001).

They conclude that the insulating (metallic) phase emerges in a striped pattern determined by surface step morphology. They consider the results illustrate the temperature behavior to be intrinsic to the material albeit 'set' by the steps. They suggest that the surface localization along with the strong determining affect of the step morphology may allow for the future local control of the MIT itself.

Employing this technique to this type of problem in particular the NNO MIT is novel and somewhat revealing. The validity of the data is unquestionable.

While the claims are robust, valid and reliable, they are not far reaching. There is little in the way of rigorous understanding of new physical phenomenon hinted at in this script.

I feel that there is further analysis to be done and a more rigorous approach would possibly allow the authors the opportunity to delve deeper into the interplay between the lateral confinement induced by the step edges and the natural out-of-plane confinement due to the surface itself with the spatial - temperature dependence of the local MIT - along with the transition process itself, i.e. the post-nucleation growth character.

In conclusion, I don't believe this work as is, is ready for publication in Nature Communications. I do however believe that a more in-depth rigorous analysis would reveal richer physics more deserving of publication.

Reviewer #3 (Remarks to the Author):

This paper reports beautiful submicron PEEM images of phase separation in heteroepitaxial nickelates as a function of temperature. The observed phase separation is dictated by the morphology of the the NdGaO₃ substrate. The authors claim in the abstract that "Our data provides new insights into the MIT in nickelates and points to a rich, nanoscale phenomenology in this strongly correlated materials," and conclude that "Our measurements point towards a strongly interconnection between structural and electronic degrees of freedom in rare-earth nickelates, suggesting a new approach for controlling phase separation at the nanoscale."

I believe that this data surely needs to be published, especially the results shown in Fig. 4, where there is an indication that the bulk of the film may be different than the surface. This is always a fundamental issue when analyzing the physical properties of thin films, are the properties including structure uniform through out the film. But I do not believe this paper should be published in Nature Communications, because what is reported is extrinsic not intrinsic, and will shed no light on the origin of MIT in correlated electron materials.

The authors seem to confuse morphology with structure. What is need is the atomic structure at the interfaces, near a step edge and away from the step edge. This structure could be different resulting in the images recorded.

I do not understand the shifts in the XAS spectra shown in Fig. 2A and there is no explanation in the

paper. My understanding is that the XAS Ni L3 spectra just becomes sharper and better defined in the insulating phase (Phase transitions 81,729, *app. Phys. Lett.* 96,233110). Is the data in Fig. 2A a result of only looking at the surface because the measurements use electron yield.

It appears that other groups report a wider hysteresis from transport measurement (*Nature Communications*, DOI: 10.1038/ncomms3714). This is always a sign of problems with the growth.

Reviewers' comments:

Reviewer #1 (Remarks to the Author):

I have reviewed the revised manuscript and the response letter. Most of my questions have been addressed adequately.

I can recommend the publications in Nature Communications, however, there are two more questions which should be addressed before its publication.

1. RSM scan in qx-qz plane does not confirm that the film is single domain. One needs to scan in qx-qy plane (e.g. Ref. 13 of revised manuscript). Without addressing this issue, it is not possible to exclude that the observed PEEM features are related to the MIT in different domains.

2. Minor question/suggestion: The PEEM is measured in TEY mode (which is surface sensitive) and XAS data is in TFY (bulk sensitive). Therefore, it is essential to compare data taken with the same detection mode, hope the authors will have an opportunity to double check their XAS data in the TEY mode to make the data consistent.

Reviewer #3 (Remarks to the Author):

I am satisfied with the response of the authors to the three reviews. The manuscript is significantly improved and should be published.

REVIEWERS' COMMENTS:

Reviewer #1 (Remarks to Authors)

Referee report for NCOMMS-16-01232B

I have gone through the response letter and revised version of the manuscript. In my earlier review letter, I had asked for a reciprocal map scan in the q_x - q_y plane to confirm that the film is single domain. The author had showed in the Supplemental a reciprocal space mapping around $(-1\ 0\ 3)$ reflection, i.e. in q_x - q_z plane.

Now, the authors responded that they had a *typographical error* in y-axis of that Figure and re-labeled it as a q_x - q_y scan in newly updated Supplemental information. However, the reflection is still $(-1\ 0\ 3)$ while q_y axis range from 2.98 to 3.10. If it is truly the q_x - q_y scan, q_y should be around 0 and not around 3(!?). The authors should clarify what the actual scan is.

With this compulsory change, I recommend the acceptance of this manuscript for publication in Nature Communications.

Reviewer #1 (Remarks to the Author):

In the manuscript 'Striped nanoscale phase separation at the metal-insulator transition of heteroepitaxial nickelates', the authors have applied PEEM technique around the first order metal-insulator transition of NdNiO₃ thin film. The authors claim to observe nucleation of the insulating phase within metallic matrix by using PEEM.

First, I would to point out that PEEM has been successfully used before to study the phase nucleation process in other systems e.g. VO₂ (arxiv:1503.07892), FeRh arXiv:1405.4319. Moreover, the existence of a mixed electronic phase within the hysteresis region of NdNiO₃ had been demonstrated before (e.g. Journal of Applied Physics 108, 063503 (2010); Journal of Physics: Condensed Matter 21 (48), 485402 (2009); Journal of Physics: Condensed Matter 21 (18), 185402 (2009)). Nevertheless, the application of PEEM to microscopically investigate the nucleation process across the 1st order MIT is new and interesting.

Before the final decision is made the author should address several questions and issues states below:

Q.1a

First, the author must add the color scale in Fig. 2(b), (c) and Fig. 3(a)! Otherwise, it is not clear how much the regions are different.

A.1a

Following the referee suggestion, we added a colour scale indicating the colour of the 2 phases. We note, however, that the PEEM data is in false colour map as the images have a strong binary character. As no intermediate states are allowed in the phase transition of the material, in fact, each region is either metallic or insulating. In this context, the intermediate colours in the PEEM maps could be related to the spatial resolution limit of the instrument.

Q.1b

The PEEM images are constructed by using the difference of area scan with photon energy of 852.0 and 852.7 eV. It is not clear what microscopic parameter do these two specific energies correspond to? It is well known that the amount of peak splitting (roughly 852.0 eV and 853.7 eV in Fig. 2(c)) can be modeled as a measure of charge transfer energy (e.g. Phys. Rev. B **83**, 161102(R) (2011)). However, as the peak splitting and the relative weight of the two peaks of L3 edge evolve with the temperature, why do the authors map the area with the same energy at all temperatures?

A.1b

The referee observation is correct, and ideally we should have repeated the analysis in fig. 2c for all the temperature points in order to obtain the optimal PEEM contrast at all temperatures. However, such analysis requires a very long measurement time not possible at a synchrotron user facility and, for practical reasons, we decided to use at all temperatures the two energies that maximise the contrast at 140K. Furthermore, we note that the choice of different pairs of energies would only affect the imaging contrast, but not change the content of the PEEM image itself, that is the spatial distribution of insulating and metallic areas. In this context, our fixed-energies choice did not affect the validity of the measurements. The maximum loss in PEEM contrast determined by our choice, in fact, is of about 50%, which is still acceptable to map the phase transition at every stage of its evolution. This choice allowed to significantly increase the number of data points along

the temperature cycles.

To explain this aspect in more detail, we added the supplementary figure S3 and the following sentence in the main text *'The use of fixed energy values slightly reduces the PEEM contrast in the high temperature region, but allowed us to perform faster acquisitions, thus increasing the number of data points taken during the temperature ramps. Further details are provided in the supplementary information'*.

Q.1c

The authors claim that the insulating phases nucleate along the terraces on the substrate by performing AFM and PEEM with the same orientation of the sample. However, there is no experimental evidence that the film used for the study is crystallographically a single domain sample. This needs to be explained.

A.1c

Our samples are single crystal epitaxial thin films. To clarify this point, we added in the supplementary S1 a reciprocal space map of the sample discussed in the main text. The reciprocal space map shows that the NdNiO₃ thin film is coherently oriented to the NdGaO₃ substrate lattice. In addition, we used atomic force microscopy (AFM) to thoroughly scan the sample surface and we found that the terraces stay coherently oriented within +/-5 degrees over the whole sample area (5x5m²). These considerations allowed us to link the orientation and dimension of the insulating domains with the sample terraces.

We added the following sentence to mention this explicitly: *"Our ex-situ comparison is allowed by the single-crystal nature of our samples (see additional XRD characterisation in the supplementary information), where the surface terraces orientation is preserved over millimetres."*

Q.1d

Can the author comment why does the insulating phase nucleate along the terraces?

A.1d

This is the first report of such behaviour on this material, and we provide a hypothesis on its origin. Our results are in agreement with reports of strain tuning of the phase transition [Scherwitzl et al., Adv. Mat. 22, 5517 (2010)] which indicates that nickelates are very sensitive to local strain fields. In this picture, the periodic strain field determined by the step and terrace surface morphology can confine the insulating phase, limiting its expansion.

We added the following sentence in the related paragraph to mention this more explicitly: *"This is consistent with reports of nickelates sensitivity to strain, suggesting that the local periodic strain field at the step edges can confine the insulating phase on the terraces, limiting its expansion."*

Q.1e

Do the author observe any time dependence in PEEM image at a fixed temperature within the hysteresis region. Earlier time dependence measurement demonstrated that Hall coefficient (Appl. Phys. Lett., 103, 182105 (2013)) drifts with time, naturally implying the metallic and insulating regions evolve in time.

A.1e

The study of time-dependence of the insulating domains evolution the referee suggests is certainly very interesting. Unfortunately, in our setup it was not possible to stabilise the temperature for a long time. As indicated in the methods section, the experiment was thus performed with slow temperature ramps of 0.5K/min. This ensured the temperature variation during a single image acquisition was negligible compared to the domains evolution. Since the single image acquisition took about 20s, we have an error of 0.2K in each data point. This limit prevented us to monitor the domains evolution over a long time. We now discussed this more clearly in the methods section with: "As each PEEM acquisition required about 20s, we estimate an error of 0.2K on each data-point. Such temperature variation was negligible compared to the phase transition evolution."

Q.1f

The width of the hysteresis curves of PEEM is much narrower compared to both the hysteresis of transport data and hysteresis of FY XAS. The authors assign this effect to the surface sensitivity of PEEM. However, the main point of this manuscript is to map out the phase nucleation around the first order metal insulator transition (existence of such nucleation process is already known) as the bulk property - as such this discrepancy must be addressed properly.

A.1f

As the referee points out, nucleation processes are a well-known characteristic of first order phase transitions, thus including the case of NdNiO₃ MIT. However, the nature of the phase separated state has never been directly observed and has just been inferred by indirect macroscopic measurements. The main result of our work is to provide evidence of a relationship between surface morphology and domains shape and distribution. These results may have a broader relevance than the specific case of NdNiO₃, suggesting a more general behaviour of MIT in oxides thin films, as related to local defects arrangement or periodic strain field.

Regarding the observed discrepancy between the hysteresis of PEEM compared to transport and FY XAS, the confinement to the surface of the observed domains is just a possible explanation. As the previous discussion in the main text could have been misleading, we now indicate more clearly how different effects can come into play in determining the observed discrepancy. The discussion follows:

"We consider two different explanations for the measured discrepancy between PEEM and transport/XAS. A possibility is that the observed domains do not fully penetrate through the whole film but are, instead, confined to the surface. In this case, our measurements indicate that the MIT at the surface occurs at a lower temperature than in the bulk, eventually related to local lattice distortions at the free boundary (i.e. surface ramplng). Another explanation involves the possible local metallisation of the material due to X-rays illumination. This is a well-known open issue when irradiating oxide materials with intense X-rays, and changes in metal-insulator characteristics have previously been reported [37]. Since the use of X-rays is an intrinsic requirement of the PEEM technique, further insight into this question will be provided by additional experiments, such as transport measurements at the nanoscale and scanning probe techniques."

Q.1g

Can the author also plot the hysteresis from TEY XAS?

A.1g

Unfortunately, we only have acquired the XAS in FY, which has been presented in figure 5.

Q.1h

One more remark: The author should carry out the PEEM experiments on a sample with the thickness comparable with to the probing depth of PEEM.

A.1h

We appreciate the remark of the referee and we believe that a study regarding the thickness-dependence of the MIT is a very interesting point and constitutes a natural follow-up of our work. However, we notice how several studies indicate that NdNiO₃ MIT is strongly dependent on thickness, in particular in the ultra-thin film limit [Scherwitzl et al. Adv. Mater. 22 (48), 5517 (2010)]. Thus, an analogous experiment on a few nanometre thick film would not be directly comparable with the present study. Furthermore, these experiments are very time-consuming and involve the use of large scale facilities, where the access to users is generally limited in time. We thus have to leave this question open to future studies.

Minor issues:**Q.1i**

Since NdNiO₃ is orthorhombic, the author should explicitly mention that the orientation (001) is in pseudo cubic setting (?)

A.1i

We used the pseudo-cubic notation throughout the whole paper but did not explicit mention that. We thank the referee for pointing out that this might be misleading, and we now added a subscript throughout the manuscript to mention this explicitly.

Q.1j

In the introduction, the author mentioned various techniques except for PEEM that have been used to map mixed metallic and insulating phases. As PEEM has been also used to investigate such phenomenon, the author should cite the appropriate PEEM refs, e.g. mentioned earlier in this report. Before taking the final decision I would like to look at the author response to my comments and the revised text.

A.1j

We thank the referee for indicating us these interesting papers that we missed. We now better discuss previous PEEM experiments in our introduction and added the citations [Y. Choi et al. arXiv:1405.4319 (2014)] and [D. Siegel et al. Physical Review B 92, 125421 (2015)].

Reviewer #2 (Remarks to the Author):

The authors present results of a temperature dependent XAS-PEEM study investigating the spatial evolution of the metal / insulator domain behaviour through the MI transition of the rare earth NdNiO₃ film grown on STO(001). They conclude that the insulating (metallic) phase emerges in a striped pattern determined by surface step morphology. They consider the results illustrate the temperature behaviour to be intrinsic to the material albeit 'set' by the steps. They suggest that the surface localization along with the strong determining affect of the step morphology may allow for the future local control of the MIT itself. Employing this technique to this type of problem in particular the NNO MIT is novel and somewhat revealing. The validity of the data is unquestionable.

Q.2a

While the claims are robust, valid and reliable, they are not far reaching. There is little in the way of rigorous understanding of new physical phenomenon hinted at in this script. I feel that there is further analysis to be done and a more rigorous approach would possibly allow the authors the opportunity to delve deeper into the interplay between the lateral confinement induced by the step edges and the natural out-of-plane confinement due to the surface itself with the spatial - temperature dependence of the local MIT - along with the transition process itself, i.e. the post-nucleation growth character. In conclusion, I don't believe this work as is, is ready for publication in Nature Communications. I do however believe that a more in-depth rigorous analysis would reveal richer physics more deserving of publication.

A.2a

Following the referee suggestion, we improved our data analysis. This resulted in the addition of 3 subfigures in the main text (figs. 3b, 3c, 4c), the consequent splitting of figure 3 in figure 3 and 4 and the addition of supplementary material which better clarifies our data analysis. We thank the referee for pointing to our attention the richer physics that was still hidden in our data, and we now feel the paper provides a deeper understanding of the nickelates MIT phenomenology.

- We evidenced the asymmetry between the metal-to-insulator and insulator-to-metal phase transition by computing FFT analysis of the PEEM images acquired during the temperature cycle. The following discussion has been added in the main text:
"A clear asymmetry between the metal-to-insulator and insulator-to-metal transition is evidenced when considering the 2D Fourier transform of the PEEM images acquired during the temperature cycle. The temperature dependence of the FFT power spectrum line-cuts along the direction perpendicular and parallel to the striped domains is reported in figs. 3b and 3c. The intensity at $k=0$ corresponds to the domains area coverage in the inset of fig. 3a, and its maximum value at 140K is used to normalise the spectrum. In the perpendicular direction (fig. 3b) we see the appearance of an intense peak, corresponding to a periodicity of about 230nm, at the nucleation of the insulating phase. This value matches with the average terrace width of fig. 1a, highlighting the direct relationship between insulating domains and surface morphology. During the cooling ramp, a longer range ordering of the domains is evidenced by the formation of higher order peaks corresponding to multiple integers of the terrace width. These features disappear when the domain area coverage reaches saturation. We point out how these additional peaks are absent during warming (dashed ellipse in fig. 3b), indicating that a different pattern underlies the disappearance of the insulating phase. The formation of the insulating domains is thus a nucleation and growth process, while their disappearance is a homogeneous melting that originates from

the domain edges. This is consistent with previous reports [25], where a supercooling mechanism was associated with the metal-to-insulator transition only. In the parallel direction (fig. 3c), instead, negligible ordering is observed, where dim peaks one order of magnitude weaker than in the perpendicular case appear. We relate this signal with the average on-terrace distance of the residual metallic matrix in the insulating phase.”

- We also performed additional data processing showing, quite surprisingly, that the spatial distribution of T_{IM} can be predicted from a single PEEM image at 140K. In particular, we added the following description:
“We find in fig. 4b that the spatial distribution of insulator-to-metal transition temperature is related to the size and shape of the domains themselves. In particular, the cores of bigger domains show higher values of T_{IM} . This indicates that the melting process of the insulating phase starts from the domain edges. To support this observation, we evaluate in fig. 4c the amount of insulating region neighbouring each insulating point in a radius of 100nm at 140K. We note how the data in fig. 4c is evaluated from a single PEEM image at 140K, in contrast to figs. 4a and 4b which are extracted by using all the images in the temperature cycle. The striking similarity between figs 4b and 4c is a clear indication of how the insulator-to-metal transition progresses continuously from the edges to the core of each domain, so that the bigger ones are the last to disappear. This is in agreement with previous reports of an intrinsic asymmetry in the phase transition of NdNiO₃ [25].”

Additional material added in the supplementary information supports our data and provides more detailed explanations for some crucial points of our analysis.

Reviewer #3 (Remarks to the Author):

This paper reports beautiful submicron PEEM images of phase separation in heteroepitaxial nickelates as a function of temperature. The observed phase separation is dictated by the morphology of the the NdGaO₃ substrate. The authors claim in the abstract that "Our data provides new insights into the MIT in nickelates and points to a rich, nanoscale phenomenology in this strongly correlated materials," and conclude that "Our measurements point towards a strongly interconnection between structural and electronic degrees of freedom in rare-earth nickelates, suggesting a new approach for controlling phase separation at the nanoscale." I believe that this data surely needs to be published, especially the results shown in Fig. 4, where there is an indication that the bulk of the film may be different than the surface. This is always a fundamental issue when analyzing the physical properties of thin films, are the properties including structure uniform through out the film.

Q.3a

I do not believe this paper should be published in Nature Communications, because what is reported is extrinsic not intrinsic, and will shed no light on the origin of MIT in correlated electron materials.

A.3a

We appreciate that the referee finds our paper interesting and that he thinks it should certainly be published. We feel Nature Communications is a very appropriate journal for our work because:

- we provide a deeper understanding of the MIT in nickelates, highlighting the direct link between sample local morphology and material properties. This is usually a very important point, in particular when considering nano-devices, for which nano-scale variations of material electronic properties are crucial
- novel perspectives for nanoscale transport and spectroscopy studies are provided
- our work will be of interest to the broad communities working on nickelate-based materials, complex oxides electronics, nanoscale imaging and synchrotron techniques, with focus on surface sensitivity

With respect to the fact the reported effect is extrinsic, since it is mediated by the interaction with the substrate, we point out how most of the work on nickelates involves thin film samples due to severe thermodynamic limitations on the growth of sizable single crystals. Therefore, the physics reported here is relevant for the vast majority of studies performed on real samples and cannot be considered extrinsic. Furthermore, thin films are the best platform to fabricate geometries to perform quantitative analysis on materials properties (i.e. Hall bars). Understanding how the material characteristics are affected by the substrate is thus of fundamental importance for further developments. We underline that our results shed light on how local strain fields related to the surface (which represents a crystal-vacuum discontinuity) or to the substrate (step-edges) can control the MIT of the material, which is often a fundamental issue when considering nano-structures and devices. For example, in literature understanding the link between structural domain walls of the SrTiO₃ substrate and enhanced conductivity in LaAlO₃/SrTiO₃ heterostructures (10.1038/NMAT3753) proved a crucial discovery in the complex oxides field.

Q.3b

The authors seem to confuse morphology with structure. What is needed is the atomic structure at the interfaces, near a step edge and away from the step edge. This structure could be different resulting in the images recorded.

A.3b

As discussed in the text, we argue that the observed effects are related to the step and terrace morphology, where the characteristic length-scale is the distance between the unit-cell steps. To avoid any possible ambiguity, we substituted throughout the text the expression “domain structures” with “domain configuration”.

Q.3c

I do not understand the shifts in the XAS spectra shown in Fig. 2A and there is no explanation in the paper. My understanding is that the XAS Ni L3 spectra just become sharper and better defined in the insulating phase (Phase transitions 81,729, *app. Phys. Lett.* 96,233110).

A.3c

The energy shift of the XAS peak position during the MIT originates from the partial change in valence state of Ni atoms in the insulating phase (10.1103/PhysRevLett.115.036401). This effect determines an increase in the L3 multiplet splitting, thus lowering in photon energy the left-most peak. We added figure S3 in the supplementary information to show this shift in more detail and included in the text the explanation: “This is consistent with an increased energy splitting of the Ni L3 multiplet in the insulating phase due to a partial change of Ni valence state [13,35]”.

Q.3d

Is the data in Fig. 2A a result of only looking at the surface because the measurements use electron yield.

A.3d

Yes, the data in fig 2a is measured in total electron yield and is therefore sensitive to the surface of the sample as stated at the beginning of the “Imaging contrast mechanism” paragraph with “*We use linear- σ X-ray polarisation and acquire the signal in total electron yield (TEY), thus probing the material surface down to a few nanometres.*”

Q.3e

It appears that other groups report a wider hysteresis from transport measurement (*Nature Communications*, DOI: 10.1038/ncomms3714). This is always a sign of problems with the growth.

A.3e

We carefully checked the paper suggested by the referee (*Nature Communications*, DOI: 10.1038/ncomms3714) and compared their transport measurement in figure 4 (+1.4%, corresponding to an NdGaO₃ substrate, which is the same used in our study) with the measurement we reported in fig 1b. Our sample shows a steeper MIT with a wider hysteresis loop and a resistivity in the metallic phase one order of magnitude lower (our is 10⁻⁴ Ohm*cm, while in the indicated study it is 10⁻³ Ohm*cm). As suggested by the referee, these differences point to a better quality of our sample. This is confirmed by the extensive characterisation we have

performed by means of X-ray diffraction (see fig. 1b in the main text and the reciprocal space map in the supplementary).

Reviewer #1 (Remarks to the Author):

I have reviewed the revised manuscript and the response letter. Most of my questions have been addressed adequately.

I can recommend the publications in Nature Communications, however, there are two more questions which should be addressed before its publication.

Q1

RSM scan in qx-qz plane does not confirm that the film is single domain. One needs to scan in qx-qy plane (e.g. Ref. 13 of revised manuscript). Without addressing this issue, it is not possible to exclude that the observed PEEM features are related to the MIT in different domains.

A1

We thank the referee for pointing out the typo in our reciprocal space map which is actually in qx, qy coordinates, thus demonstrating the single domain nature of our sample. This is consistent with the results of Ref 13, as pointed out by the referee. We corrected the y-axis label of the figure S1 in the supplementary information.

Q2

Minor question/suggestion: The PEEM is measured in TEY mode (which is surface sensitive) and XAS data is in TFY (bulk sensitive). Therefore, it is essential to compare data taken with the same detection mode, hope the authors will have an opportunity to double check their XAS data in the TEY mode to make the data consistent.

A2

We agree with the referee's suggestion that this could be an interesting comparison, but unfortunately we do not have temperature-dependent XAS data taken in TEY mode. We are aware that such data would have allowed a better understanding of the difference between the XAS FY and PEEM and we regret its absence. Nonetheless, the purpose of fig.5 is to mark the different temperature dependence of the metal-insulator transition when measured with surface or bulk-sensitive techniques. This suggests that further experiments are required to understand the relationship between surface and bulk insulating domains.

Reviewer #1 (Remarks to the Author):

I have gone through the response letter and revised version of the manuscript. In my earlier review letter, I had asked for a reciprocal map scan in the qx-qy plane to confirm that the film is single domain. The author had showed in the Supplemental a reciprocal space mapping around (-1 0 3) reflection, i.e. in qx-qz plane.

Q1

Now, the authors responded that they had a typographical error in y-axis of that Figure and re-labeled it as a qx-qy scan in newly updated Supplemental information. However, the reflection is still (-1 0 3) while qy axis range from 2.98 to 3.10. If it is truly the qx-qy scan, qy should be around 0 and not around 3(!?). The authors should clarify what the actual scan is.

With this compulsory change, I recommend the acceptance of this manuscript for publication in Nature Communications.

A1

To clarify our analysis we performed additional XRD measurements. To avoid any possible confusion with axes labels and reference systems, we report all the results in pseudocubic notation, where h,k,l denote the reciprocal lattice directions $(100)_{pc}$, $(010)_{pc}$ and $(001)_{pc}$. The aligned phi--scan measurements of the substrate and thin film confirm a crystallographically aligned heteroepitaxy. Reciprocal space maps in the h,l and k,l planes confirm the thin film is coherently oriented to the substrate lattice.